# Hollow Fiber Membrane for Organic Solvent Nanofiltration: A Mini Review

**DOI:** 10.3390/membranes12100995

**Published:** 2022-10-13

**Authors:** Liyang Liu, Shaoxiao Liu, Enlin Wang, Baowei Su

**Affiliations:** 1Key Laboratory of Marine Chemistry Theory and Technology, Ocean University of China, Ministry of Education, 238 Songling Road, Qingdao 266100, China; 2College of Chemistry & Chemical Engineering, Ocean University of China, 238 Songling Road, Qingdao 266100, China

**Keywords:** organic solvent nanofiltration (OSN), hollow fiber membrane, review

## Abstract

Organic solvents take up 80% of the total chemicals used in pharmaceutical and related industries, while their reuse rate is less than 50%. Traditional solvent treatment methods such as distillation and evaporation have many disadvantages such as high cost, environmental unfriendliness, and difficulty in recovering heat-sensitive, high-value molecules. Organic solvent nanofiltration (OSN) has been a prevalent research topic for the separation and purification of organic solvent systems since the beginning of this century with the benefits of no-phase change, high operational flexibility, low cost, as well as environmental friendliness. Especially, hollow fiber (HF) OSN membranes have gained a lot of attention due to their high packing density and easy scale-up as compared with flat-sheet OSN membranes. This paper critically reviewed the recent research progress in the preparation of HF OSN membranes with high performance, including different materials, preparation methods, and modification treatments. This paper also predicts the future direction of HF OSN membrane development.

## 1. Introduction

A great quantity of organic solvents are employed in industrial production, which could take up 80% of the total chemicals used in pharmaceutical and related industries [1]. However, according to statistics, the reuse rate of organic solvents in pharmaceutical industry is less than 50% [1]. Compared with the conventional separation methods of organic solvent systems such as distillation and evaporation, organic solvent nanofiltration (OSN) technology has received much attention due to its low energy consumption, ease of operation, and environmental friendliness [2,3]. Researchers have demonstrated the feasibility of OSN technology in many industrial productions, such as the recovery of active pharmaceutical ingredients (APIs) [4,5], recovery of solvents from crystalline mother liquors [6], solvent recovery and dewaxing of lube oil [7], biorefineries [8], natural product isolation [9], organocatalysis [10], solvent exchange [11], etc. At the beginning of the 21st century, there was already a large plant using OSN technology to recover organic solvents from lube oil with a maximum feed rate of 72,000 barrels per day [7].

The key to OSN technology is OSN membranes. According to the membrane configurations, OSN membranes could be classified as flat sheet (FS) [12,13], tubular [14], or hollow fiber (HF) [15] OSN membranes (as shown in Figure 1). Among the three configurations, HF OSN membranes show greater potential in practical application of OSN technology due to their advantages of high packing density, self-supporting structure, and low fouling tendency [16,17]. Recently, researchers have carried out many studies on the HF OSN membranes to improve their solvent resistant [18], mechanical strength [19] and separation performance [15]. 

To date, there are several published review articles which mainly focus on the recent progress of OSN membranes [21,22,23] and on the research progress and application of HF nanofiltration membranes [16,17,24]. However, a comprehensive review about HF OSN membranes is still lacking. This review aims to summarize the research progress of HF OSN membranes, point out the lack of current research, and further predict the development direction of HF OSN membrane technology.

## 2. Fundamental

This section will briefly discuss the OSN process, separation metrics, spinning methods, and performance metrics of HF OSN membranes.

### 2.1. OSN Processes

As shown in Figure 2, the OSN process is a pressure-driven membrane separation process which is used for the recovery of molecules with molecular weight between 50 and 2000 Da [25]. As shown in Figure 2, pressure forces small molecules to pass through OSN membranes, and large molecules are rejected. The OSN process could be divided into three categories: concentration (i.e., separating a single solute from solvents), solvent exchange (i.e., separating solvent from another solvents), and purification (i.e., separating two or more solutes) [2,16]. Usually, the OSN process is operated at room temperature [26] and under a pressure lower than 30 bar [21,26,27].

Although there are many published works about OSN membranes, most of the current research on OSN process is focused on the concentration process, which is operated at room temperature [21], and there are fewer published reports on emerging OSN processes such as the separation of non-polar solvent systems [28], fractionation [8,29,30,31], and high-temperature OSN process [32,33].

**Figure 2 membranes-12-00995-f002:**
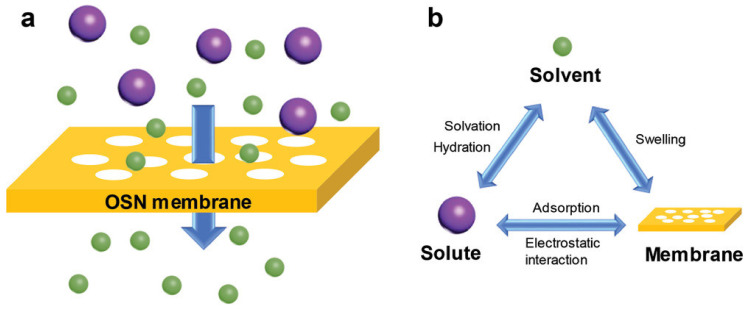
(**a**). Schematic representation of sieve effect based OSN membrane separation. (**b**). Interactions between membrane-solvent-solute. Reprinted and adapted with permission from Ref. [34]. Copyright 2020, John Wiley and Sons.

### 2.2. Separation Mechanisms

Nanofiltration membranes applied to aqueous systems cannot be directly applied to organic solvent systems since the properties of organic solvents are different from those of water, which makes the separation mechanisms of aqueous systems not directly applicable to organic solvent systems. As an example, the charge effect applied to aqueous systems is not applicable to organic solvent systems, since the charge effect therein is negligible [35]. OSN membranes are supposed to work mainly based on sieve effect [2]; however, factors such as the properties of the solute, solute–membrane interactions, and solvent–membrane interactions could affect the OSN separation performance of OSN membranes [36], and there is little research focused on these fundamental aspects [37].

### 2.3. Preparation of Substrates

Like FS OSN membranes which are more studied, HF OSN membranes are mainly made of polymers materials because of the advantages of polymers such as low cost and ease of manufacture and handling [38]. Most substrates of HF OSN membranes are prepared via non-solvent induced phase separation (NIPS) method or thermally induced phase separation (TIPS) method. The NIPS method is more frequently used as it does not need to spin fibers at high temperature or in other specific conditions [18]. Dry-jet wet-spinning is more widely used among the NIPS methods, the basic process of which is pushing dope solution and bore fluid (i.e., internal coagulation bath) co-currently entering a spinneret with an external driving force; subsequently, the extruded fibers pass through a certain air gap into a coagulation bath and form hollow fiber substrates [39]. Figure 3 shows the schematic diagram of the spinning process for hollow fiber substrate.

Figure 4 shows the schematic depiction of a hollow fiber membrane fabrication system.

### 2.4. Performance Metrics

The main performance parameters for evaluating HF OSN membranes are solvent permeation and solute rejection. To ensure practical industrial applications, HF OSN membranes need to work stably in organic solvent systems with high solvent permeance and solute rejection. 

Solvent permeance is defined as the volume of solvent permeated through unit membrane area within unit time interval under unit transmembrane pressure difference, it is calculated according to Equation (1), and its commonly used unit is L m^−2^ h^−1^ bar^−1^ or L m^−2^ h^−1^ MPa^−1^.
(1)$$P=\frac{\Delta V}{A\cdot \Delta t\cdot \Delta P}$$
where *P* (L m^−2^ h^−1^ bar^−1^) is the permeance of organic solvent, Δ*V* (L) is the volume of solvent permeated during time interval Δ*t* (h), Δ*P* (bar) is the transmembrane pressure difference, *A* (m^2^) is the effect area of the membrane.

Solute rejection (*R*) reflects the selectivity of HF OSN membrane and is calculated according to Equation (2):(2)$$R=\frac{C_{F}-C_{P}}{C_{F}}\times 100\%$$
where *R* represents the solute rejection, *C*_F_ represents the solute concentration in feed solution, and *C*_P_ represents the solute concentration in permeate solution.

Usually, dyes are widely used as solutes in OSN filtration experiments since the molecular weights of most dyes are in the separation range of OSN, and the concentration of dyes could be directly calculated by their absorbance measured by ultraviolet–visible spectroscopy at specific wavelengths, without the necessity of complex instruments [21]. 

## 3. Structures and Preparation Methods of HF OSN Membranes

Currently, polymers are the most used materials for preparing HF OSN membranes, and polyimide is the most used polymeric membrane material, and NMP, THF, DMF, and DMAc are the common solvents used for preparing dope solutions. The NIPS method is the most used method for preparing hollow fiber substrates and crosslinking the nascent substrates by using diamines is the most common post-treatment method. For the IP process, amines are the commonly used aqueous phase monomers, and hexane is the commonly used organic phase solvent. According to their structure, HF OSN membranes could be classified as integrally skinned asymmetric (ISA) HF OSN membranes and composite HF OSN membranes according to their structure [16,25].

### 3.1. ISA HF OSN Membranes

ISA HF OSN membranes refer to membranes in which the top dense separation layer and the support layer are of the same material [16]. The separation layer of ISA HF OSN membrane was prepared via phase inversion method. The advantages of ISA HF OSN membranes are relatively simple preparation, relatively simple backwashing of fouled membranes, and high mechanical strength [21]. However, compared with composite membranes, ISA HF OSN membranes are usually more difficult to regulate the formation of pores and separation layers, and their separation performance is lower [41,42]. Figure 5 shows the surface and cross-sectional morphologies of a typical ISA HF OSN membranes which has 939 nm thickness dense outer selective layer.

According to the structure of HF OSN membranes, ISA HF OSN membranes could be classified into nanomaterial-free integrally skinned asymmetric (NISA) HF OSN membranes and HF OSN mixed matrix membranes (MMMs).

#### 3.1.1. NISA HF OSN Membranes

The formation process of HF OSN membranes differs from that of FS OSN membranes as the existence of non-solvent phase in the inner lumen of fibers, which causes the phase inversion process to happen on both the inner and outer surfaces of the HF substrates. Early research works in ISA HF OSN membrane preparation focused on the effects of membrane materials and spinning conditions on membrane morphology [43,44], and how to make membranes solvent-resistant [45]. 

Darvishmanesh et al. [43] studied the effect of spinning parameters (e.g., the polymer content in the dope solution and the ratio of solvent to non-solvent in the bore fluid) on the morphological changes of the resultant ISA HF OSN membranes, and concluded that their morphology could be optimized and their ISA HF OSN performance could be improved by adjusting the spinning parameters (e.g., polymer concentration, dope fluid flow rate, bore fluid flow rate, and bore fluid composition). Loh et al. [44] spun polyaniline (PANi) HF substrate by dry-jet-wet-spinning (also known as gel spinning) PANi dope solution with maleic acid added as a pore-forming agent, where the mass fraction of maleic acid added was half of the PANi mass fraction. They then immersed the HF substrate in an alkaline solution to remove the residual organic acids, and then heat-treated the substrates at 180 °C for one hour to obtain PANi HF OSN membranes. OSN performance testing results showed that the prepared membrane was stable in acetone with a MWCO of 350 Da. For comparison, fibers spun without the addition of maleic acid as a pore-forming agent had little or no solvent permeance. Their work demonstrated the importance of the pores for solvent transport and that organic acids could be doped as pore-forming agents in preparing HF OSN membranes, which provided a more feasible method for the development of HF OSN membranes. However, the solvent permeance of the HF OSN membranes they prepared (pure acetone permeance of 1.53 L m^−2^ h^−1^ bar^−1^) still needs to be enhanced.

To simplify the membrane manufacture process, Dutczak et al. [45] added poly(ethylene imine) (PEI) crosslinker to the bore fluid to achieve the spinning and crosslinking process in one step. The results of the organic solvent immersion experiments showed that the mass loss ratio of the crosslinked P84 HF OSN membranes prepared under the optimal conditions was 20 ± 3% after 11 days of immersion in the nonprotonic polar solvent N-methyl-2-pyrrolidone (NMP). In contrast, the uncrosslinked HF OSN membranes were almost completely dissolved in NMP. The crosslinked HF OSN membranes prepared under the optimal membrane production conditions increased the toluene permeance from 0 to 1.5 ± 0.1 L m^−2^ h^−1^ bar^−1^ compared with the corresponding uncrosslinked membranes. Their work demonstrated the feasibility of combining the spinning and crosslinking steps, which simplified the membrane manufacturing process and provided a new idea for regulating the structure of the prepared HF OSN membranes. However, the prepared HF OSN membranes could not work stably in NMP because of their much high mass loss ratio (i.e., 20 ± 3%) when immersed in NMP. Their subsequent work [46] indicated that the HF OSN membrane could be crosslinked to different degrees by adjusting the crosslinking conditions (e.g., composition and temperature of the crosslinker solution) and thereby regulating the separation performance of the crosslinked HF OSN membrane. 

To enhance the mechanical properties of HF OSN membranes, Lim et al. [42] spun Torlon^®^ 4000T-MV polyamide-imide (PAI) HF substrates by using inorganic salt lithium chloride (LiCl) as the porogenic agent, and then immersed the substrates in (3-aminopropyl)trimethoxysilane (APTMS) silicone crosslinker to prepare APTMS crosslinked PAI HF OSN membranes. Energy dispersive X-ray spectrometer (EDX) and tensiometer characterization results showed that the silica-based crosslinker was uniformly distributed in the prepared PAI HF OSN membranes and successfully crosslinked the substrates to form an organic-inorganic crosslinking network structure and increased the hydrophilicity of the prepared membranes. Tensile test results showed that the tensile modulus of the crosslinked PAI HF OSN membrane increased with the crosslinking temperature or crosslinking time, but the membrane became more brittle. The crosslinked PAI HF OSN membranes worked stably in rose bengal (RB, 1017 Da) isopropanol (IPA) solution, and showed an impressive high IPA permeance of 6.4 L m^−2^ h^−1^ bar^−1^ with RB rejections of 97.3%.

In recent years, research on ISA HF OSN membranes has turned to preparing membranes with high OSN performance and developing green and environmentally friendly membrane manufacturing processes.

Some works focus on low cost and facile preparation of HF OSN membranes. Tham et al. [47] crosslinked polyacrylonitrile (PAN) HF substrates by using low cost and easy-to-obtain hydrazine as a crosslinker to produce crosslinked PAN HF OSN membranes. The prepared crosslinked PAN HF OSN membrane has an impressive high rejection (>99.9%) for Remazol Brilliant Blue (RBB, MW = 626). In their subsequent work [48], they used renewable plant bio-phenolic tannic acid to modify the hydrazine-crosslinked PAN HF OSN membranes to enhance their mechanical and separation performances. Under 3 bar pressure, the prepared membranes showed up to 100% rejection for Evans Blue (EB, 960.81 Da) in methanol. Their works provide innovative ideas for the simple and economical preparation of HF OSN membranes because hydrazine hydrate is cheap and easy to obtain. However, hydrazine hydrate is toxic, corrosive, and flammable, which limits the wide application of their membrane preparation strategy [49]. Tashvigh et al. [50] reported a method for the preparation of polybenzimidazole (PBI) HF OSN by protonation of PBI HF substrates through inorganic acid (i.e., sulfuric acid) immersion. After sulfuric acid immersion, hydrogen bonds formed between PBI and sulfuric acid strengthened the polymer network and kept the prepared PBI HF OSN membrane stable in dimethyl sulfoxide (DMSO), dimethylacetamide (DMAc) and DMF. However, the membrane is unstable in NMP because NMP weakens the hydrogen bond between PBI and sulfuric acid. Xu et al. [39] prepared P84 polyimide (PI) HF OSN membranes with a smooth surface by coating PEI on the surface of P84 PI substrates, followed by glutaraldehyde (GA) crosslinking and final 1,6-hexanediamine (had) crosslinking. The prepared HF OSN membranes showed a potential for application in solutions of moderately polar to non-polar solvents. However, the prepared HF OSN membranes showed a low EtOH permeance of 4.5 L m^−2^ h^−1^ bar^−1^. 

According to the pore-scale flow and transport mechanisms in porous media, the pore size and distribution could affect the OSN separation performance of HF OSN membranes. Some works have been carried to enhance the OSN performance of ISA HF OSN membranes by adjusting their surface parameters (e.g., pore size and distribution). Wang et al. [51] prepared a series of P84 PI HF OSN membranes by altering the spinning parameters (e.g., air gap length, bore fluid composition, and the fluid flow rate of polymer dope solution). They analyzed the effect of spinning parameters on membrane morphology by both thermodynamic and kinetic factors. Song et al. [18] studied the microscopic mechanisms regulating the separation layer of prepared membrane and explained the relationship between the molecular weight variation of the same crosslinker and the variation of membrane pore size parameters. Their work provides constructive guidance for manufacturing HF OSN modules for commercial HF OSN membrane applications. Li et al. [31] immersed crosslinked PAI HF membranes in calixarene solution for one hour and the characterization results demonstrate that the free volume and pore size could be regulated by changing the concentration of calixarene. Wang et al. [30] prepared PBI HF OSN membranes by a similar strategy. The prepared HF OSN membrane showed an impressive rejection (99.5%) for RBB (MW = 626 Da) dissolved in acetone. These works provide feasible strategies for the preparation of HF OSN membranes.

DMF, NMP, and DMAc are the common solvents used to prepare dope solution, which are both toxic to humans and to the environment. In addition, diamine, the commonly used crosslinker, is also harmful to human health and the environment [25]. Some scholars have explored green production process to prepare HF OSN membranes. Jeon et al. [52,53] spun polyamide (PA) 6 HF OSN fiber membranes in one step via TIPS method with using green dimethyl sulfone and sulfolane as non-toxic diluents. The preparation of polyamide 6 HF OSN membranes does not require the crosslinking step, which provides a unique approach to further simplify the membrane manufacturing process. Zhao et al. [54] developed a green crosslinking process using potassium persulfate (K_2_S_2_O_8_) crosslinker to prepare PBI HF OSN membranes which could be stable in organic solvent systems, and explored the method of backwashing to remove fouling from the membranes. Their work provides guidance for the green preparation of HF OSN membranes. Falca et al. [55] prepared various HF fiber membranes by dissolving cellulose in different ionic liquids in a green method, and the prepared HF membranes could work stably in ethanol system. However, their cellulose HF membranes only showed a high rejection for specific dyes (i.e., Congo Red), which indicated that these cellulose HF membranes mainly separated dyes by adsorption.

#### 3.1.2. HF OSN MMMs 

The concept of mixed matrix membranes was first introduced in the field of gas separation membranes by adding nano-fillers to membranes to break the trade-off effect of polymeric membrane materials [56]. The introduction of nano-fillers regulates the transfer phenomena occurring in the prepared membranes, thus improving their OSN performance [57]. The introduction of nano-fillers could also reduce the compaction effect (i.e., the permeance of polymer membranes would decrease to a stable value from the initial value when working under specific pressure) [58]. 

The key challenge to prepare MMMs is how to synthesize nano-fillers at low cost, and make them distributed uniformly in the prepared membranes matrix [57]. Farahani et al. [59] prepared HF OSN MMMs by doping amine-functionalized multi-walled carbon nanotubes (NH_2_−MWCNTs) into the dope solution and then crosslinked the nascent P84 PI substrate fibers to obtain NH_2_−MWCNTs/P84 HF OSN MMMs. The characterization and separation performance results showed that the introduction of NH_2_–MWCNTs improved the mechanical and separation performances of the membranes. They simulated the OSN process in food and pharmaceutical industries and got satisfactory results. Li et al. [60] dropwise added Titanium Dioxide (TiO_2_) sol into the dope solution, which contained polyamic acid, EtOH, and NMP, and spun the dope solution to hollow fibers. Subsequently, they imidized the fibers and prepared poly(4,4′-oxydiphenylene pyromellitimide) (PMDA-ODA) PI HF MMMs. Figure 6 shows the SEM observation and EDS analysis of the cross-section of the prepared PAA HF MMMs. It could be seen that the TiO_2_ nanofillers were uniformly distributed in the prepared HF MMMs. Other characterization and test results proved that the introduction of TiO_2_ accelerated the phase inversion of polymer solution, increased the pore size, and changed the surface properties of the prepared HF membranes. 

Some researchers explored the green preparation methods to prepare HF OSN MMMs. Kato et al. [61] spun polyamide 6 HF OSN MMMs containing delaminated layered montmorillonite nano-fillers in one step via TIPS process without crosslinking step. Compared with their previous work [33], inorganic nano-fillers improved the thermoplasticity and crystallinity of polyamide 6, thus further enhancing the stability of the prepared HF OSN membranes. Their work is an important guide to simplify the membrane preparation process. However, the prepared polyamide 6 HF OSN MMMs showed low solvent permeability (MeOH permeance of ~0.21 L m^−2^ h^−1^ bar^−1^), probably due to the high polymer concentration (30 wt. %) in the dope solution.

Table 1 compares the separation performance of the aforementioned ISA HF OSN membranes, which are made of various materials. Compared with PI and PAI, PBI and cellulose may be the next generation materials to prepare ISA HF OSN membranes due to their high permeability to certain solvents. 

### 3.2. Composite HF OSN Membranes

The majority of studies on HF OSN membranes are focused on ISA HF OSN membranes, and there are fewer studies on composite membranes [63]. Since the separation layer properties of ISA membranes are difficult to be precisely regulated, some studies have focused on the research of composite membranes [64] which have different separation performances to meet different commercial needs by regulating the properties of the separation layer and the support layer, respectively [65,66]. Presently, composite HF OSN membranes could be prepared via coating [67], interfacial polymerization (IP) [15,68], etc. 

#### 3.2.1. Thin Film Composite HF OSN Membranes Prepared via IP 

Thin film composite (TFC) HF OSN membranes mean HF OSN composite membranes with a thin selective layer (also called skin layer). Compared with ISA HF OSN membranes, TFC HF OSN membranes usually have better solvent permeability, and the thin-film skin layer could be individually designed to meet the separation requirements of different processes. However, TFC membranes face the problem of shedding of selective layer [69] and difficulty to avoid or remove membrane fouling [69,70]. 

The main process for preparing the selective layer of TFC HF OSN membrane is the interfacial polymerization (IP) process, which could be mainly summarized as the generation of a selective thin skin layer on the substrates by the reaction of two monomers which dissolved in two mutually insoluble solvents, respectively (shown as Figure 7). The two solvents are not mutually soluble in order to produce a stable phase interface [34]. 

Because of the self-inhibiting effect of IP process, the thickness of the skin layer generated via IP is generally small (<100 nm) [71]. A typical skin layer of TFC membranes is polyamide (PA), which is produced by the reaction between m-phenylenediamine (MPD) and trimesoyl chloride (TMC) [21,72]. Figure 8 shows the SEM images of a typical TFC HF OSN membrane with a 60 nm thickness dense PA skin layer. The key to the preparation of TFC HF OSN membranes via IP method is the precise regulation of the IP process because the high diffusion rate of MPD into the organic phase and the large kinetic constants of the IP reaction make the IP process difficult to regulate precisely, resulting in high PA crosslink-degree and low solvent permeance [34]. There are five main strategies to regulate the IP reaction process forming PA skin layer precisely, including controlling the storage of amine monomer on the substrates [73], controlling the diffusion rate of amine monomer [15], removing the heat of reaction in time [74], preventing the formation of nano-sized bubbles [74], and inhibiting the downward growth of the polyamide layer [75].

The special structure of hollow fibers makes aqueous solution difficult to be uniformly distributed on the surface of the substrates, thus making the preparation of TFC HF OSN membranes much more difficult [15,76]. Some work has been reported about forming a skin layer on the inner surface of HF substrates. Kosaraju et al. made the first attempt to prepare TFC HF OSN membranes [77]. They pre-wetted the propylene (PP) substrates with acetone and subsequently oxidized them with chromic acid solution, and then pushed the solution containing PEI and isophthaloyl dichloride into the inner lumen of the PP hollow fibers, respectively. After that, they thermally treated the prepared membranes for 20 min to obtain PP HF OSN membranes, which showed MeOH permeance of 1.47 L m^−2^ h^−1^ bar^−1^ and a rejection of 88% for BBR (MW = 826). Their work is of guidance to the development of subsequent HF OSN membranes. Goh et al. [76] attempted the PEI-TMC IP reaction by reacting different concentrations of PEI to 0.13% wt. % TMC for a long time (∼20 min) on the inner surface of P84 HF substrates. The characterization results proved that the IP reaction formed a PA separation layer with a thickness of 50–70 nm in the inner surface of the P84 HF substrates and the TFC HF membranes were successfully prepared for the low-pressure OSN process. In their subsequent work [68], they prepared HF OSN modules containing 100 fibers via a similar method to simulate industrial APIs HF OSN separation processes. The prepared HF OSN modules show 95% rejection of APIs (i.e., levofloxacin) and EtOH permeance of 2.33 L m^−2^ h^−1^ bar^−1^ in long-term testing. Their work provides a potential platform for large-scale manufacturing of HF OSN membranes. Some researchers tried to conduct a hydrophobic selective layer on the inner surface of HF substrates to meet the requirement of non-polar solution, since most HF OSN membranes were prepared for the separation and purification of polar solvent systems, which showed very low permeance for non-polar solvents, and there were few works reported about the OSN filtration of non-polar solution [78]. Gao et al. [28] successfully formed a hydrophobic covalent organic framework (COF) separation layer in situ on the inner surface of the hydrophilic crosslinked PI HF substrates. OSN performance tests showed that the prepared HF OSN membranes performed an impressive high solvent permeance to acetone of 395.21 L m ^−2^ h ^−1^ bar ^−1^, which was 130 times higher than that of commercial FS OSN membranes [28]. Their membrane preparation strategy shows enormous potential for preparing Janus-like HF OSN membranes with both hydrophilic and hydrophobic separation layers to meet the separation requirements of different solute-solvent solution.

Compared with the HF OSN membranes with the selective layer on their inner surface, the HF OSN membranes with the skin layer on the outer surface are more suitable for commercial applications due to their lower pressure drop, less tendency to fiber blockage, and larger membrane area under the same conditions (e.g., operating conditions, module preparation, and fiber size) [79,80]. However, it is more difficult to form a defect-free PA skin layer on the outer surface of HF substrates, especially in large-scale production, which hinders the commercialization of outer-selective HF OSN membranes [81,82].

Some researchers made attempts to prepare HF OSN membranes after modifying the surface of HF substrates, since the physical and chemical properties of the outer surface of HF substrates, e.g., surface pore size, porosity, and surface chemistry, could affect the morphology, thickness, and crosslink-degree of the PA selective layer formed via IP [41]. Substrates with high surface porosity, narrow pore size distribution, and moderate average pore size are suitable for the forming of high-performance PA layers [83]. To adjust the pore size and pore distribution of the substrates, Sun et al. [73] prepared PI/PI dual layer substrates with fewer surface defects, more rational pore size, and narrower pore size distribution by adjusting the components in the dope solution. Then, they prepared PA-PI-PI three-layer composite HF OSN membranes via vacuum-assisted interfacial polymerization. The results proved that the separation performance of composite membranes could be optimized by modulating the physical properties of the base membrane, e.g., pore size and pore size distribution. Though the prepared HF OSN membrane’s solvent permeance was low (MeOH permeance of 0.65 L m ^−2^ h ^−1^ bar ^−1^), their preparation method (vacuum-assisted interfacial polymerization) was of insightful guidance for the subsequent preparation of TFC HF OSN membranes.

Since long reaction times (e.g., 10 min) are unfavorable for large-scale manufacturing, and high concentrations of reactive monomers increase the cost of membrane preparation, some researchers made attempts to conduct the IP reaction on the outer surface of HF substrates using ultra-low concentration monomers within short IP duration. Su et al. [15] successfully prepared a thin (∼60 nm) PA skin layer on the outer surface of P84 PI substrates at low monomer concentration (0.05 wt. % MPD and 0.15 wt. % TMC) and interfacial polymerization time (10 s). OSN performance showed that the TFC membranes had an impressive high rejection, which was up to 100% for RDB. They then added graphene oxide (GO) nanosheets to the aqueous solution and successfully prepared P84 thin film nanocomposite (TFN) HF OSN membranes. Characterization showed that the skin layer thickness of the prepared TFN membranes was only 40 nm, which demonstrated that the addition of GO nanosheets effectively regulated the IP process. OSN results showed that the permeance of TFN membranes was increased by 65% compared with that of TFC membranes. Their work proves the feasibility of commercial production of HF OSN membranes. 

#### 3.2.2. Dual Layer HF OSN Membranes

Dual layer hollow fiber membranes offer the idea of reducing membrane production costs by using only the expensive high-performance material as the separation layer instead of the entire fiber [84]. Figure 9 shows the morphology of a typical dual-layer HF membrane which has two obviously different layers.

Due to the different properties of the two polymers, adhesion or delamination often occurs in preparing dual layer HF OSN membranes, which are negative for the dual layer HF OSN membranes’ performance [85]. The key to the preparing dual layer HF OSN membranes is the elimination of adhesion and delamination. Wang et al. [86] studied the mechanism of adhesion and delamination of dual layer membranes, and prepared non-delaminated dual layer PI/polyetherimide HF OSN membranes by adding fluoro-substituted aromatic amines. Their work provides guidance for the design and manufacture of more dual layer HF membranes. 

Because of the inherent properties of polymeric membrane materials, ISA HF OSN membranes are difficult to operate at higher pressures [51], which limits their commercial application. Wang et al. [87] prepared braid-reinforced P84 HF OSN membranes with a stable working pressure three times that of traditional P84 HF OSN membranes by coating P84 PI dope solution on high-strength polyester (PET) braid support. Zhao et al. [19] prepared braid reinforced PBI HF OSN membranes via a similar process. As shown in Figure 10, the mechanical strength of braid reinforced PBI HF OSN membranes was greatly improved compared with conventional self-supporting PBI HF OSN membranes. 

To simplify the preparation processes of HF OSN membranes and reduce the materials costs, Sun et al. [88] prepared a dual layer hollow fiber membrane, which PBI was the outer separation layer and PI was the inner support layer, by using a triple-orifice spinneret. When spinning, the bore fluid is the solution containing hyperbranched poly(ethylenimine) (HPEI) which serves as the crosslinking agent. Their work eliminates the post-spinning crosslinking step and provides a potential platform for the facile one-step preparation of HF OSN membranes. Their work shows great prospects in the application of dual layer HF OSN membranes. 

#### 3.2.3. Other Composite HF OSN Membranes

There have been some composite HF OSN membranes prepared via other methods. Mahalingam et al. [67] firstly prepared HF OSN membranes by spray coating with a 16 nm thickness GO skin layer on the outer surface of crosslinked polyetherimide substrates. Lai et al. [32] successfully prepared poly(p-phenylene terephthamide) resin (PPTA) HF OSN membranes that could work stably at 80 °C DMAc by the chemical vapor deposition (CVD) method. 

In addition to the common polymer HF OSN membranes, there have been a few published articles about ceramic HF OSN membranes. Overall, the substrates of ceramic HF OSN membranes are obtained by sintering after spinning out the HF precursors via the NIPS method. After that, the ceramic HF OSN membranes are obtained by surface modification (e.g., vacuum filtration [50,89], coating [50,89,90], and IP [33]) of ceramic substrates.

Li’s group successfully prepared GO-ceramic HF OSN membranes by vacuum filtration, which showed a MeOH permeance of 3.97 L m^−2^ h^−1^ bar^−1^ and a MO (MW = 327) rejection of 97%. However, the drying-related shrinkage led to micro-structural instability in the prepared membranes, resulting in the prepared GO-ceramic HF OSN membranes being stored only in liquid. Wang et al. [89] prepared HF nanofiltration by coating *γ*-Al_2_O_3_ sols on *α*-Al_2_O_3_ substrates as a selective layer, and the prepared HF nanofiltration membranes could be stable when immersing in organic solvents and mild aqueous media with pH ranging from 3 to 11. Lee et al. [90] prepared inorganic ceramic HF OSN membranes via similar methods. The newly developed ceramic membranes showed excellent hexane permeability of 4.3 L m^−2^ h^−1^ bar^−1^, but the HF OSN membranes may be unstable under harsh conditions (e.g., strong acid and high temperature solvents). Abadikhah et al. [50] successfully prepared TiO2@rGO TFN HF OSN membranes by interfacial polymerization after immersing ceramic substrates in polymer dope solutions. Introducing TiO2@rGO nanosheets significantly improved the permeance of EtOH but decreased the permeance of non-polar solvent (i.e., hexane). However, the complex preparation process limits the scale-up trials of the prepared membranes. Recently, Huang et al. [33] successfully fabricated PA separation skins on ceramic substrates directly via direct interfacial polymerization. The resultant membranes showed stable OSN performance in harsh conditions (e.g., strong acid, strong base and high temperature), and could be suitable for separation of harsh solvents at high temperatures. However, the surface of the prepared PA-ceramic HF membranes is rougher (Ra = 70.6 nm), which may lead to serious membrane fouling problems during long-term operation. Although some progress has been made with ceramic HF OSN membranes, their brittle characteristic, manufacturing difficulties, and high costs limit the application of ceramic HF OSN membranes [34]. In addition, the preparation of ceramic precursor solutions requires the use of environmentally unfriendly solvents [25] (e.g., NMP [33,50] and DMAc [90]). These ceramic HF OSN membranes are prepared by modifying ceramic HF substrates.

Table 2 shows the aforementioned composite HF OSN membranes’ OSN performances, which were tested by separating single solute solutions below 10 bar. Compared with ISA HF OSN membranes, composite HF OSN membranes showed higher solvent permeance and more expandability for membrane preparation. Of these composite HF OSN membranes, the HF OSN membranes with the COF selective layer showed impressive solvent permeance which was one order of magnitude higher than that of others [28].

## 4. Treatment Methods of HF OSN Membranes

After spinning, the nascent membrane fibers often require further processing to meet the requirements of OSN separation. The modification of OSN membranes could be divided into surface modification or modification of the entire membranes [91].

### 4.1. Integral Modification

#### 4.1.1. Crosslinking Treatment

Most polymer membranes would swell or dissolve when exposed to some organic solvents, especially non-protonic solvents such as DMF and THF, resulting in loss of structural integrity and separation properties [35]. Thus, general polymeric membranes need to be treated (e.g., crosslinking) to obtain solvent resistance [45].

The major treatment method to treat membranes is crosslinking, which is to improve the HF OSN membranes resistant to solvents and plasticization [92]. The present crosslinking methods used for HF OSN preparation are mainly chemical crosslinking methods [15,86]. Crosslinking treatment is usually carried out after spinning fibers, which means that crosslinking treatment is a post-treatment method and brings additional process steps. Crosslinkers could also be added to the bore fluid to achieve phase inversion and crosslinking at the same time to simplify the membrane preparation process [45,88].

There have been several special published reviews summarizing the crosslinking process [49,92]. In brief, amines are the most widely used crosslinkers [49] due to their ability to react with a wide range of commonly used polymer materials (e.g., PI [45,68,73], PBI [30,88], PAN [47,48]). Figure 11 shows the mechanism of crosslinking of amines and PI.

Table 3 shows the advantages and disadvantages of various crosslinking methods.

#### 4.1.2. Introducing Nanofillers in Substrates

Incorporation of nanomaterials into the substrate to prepare HF OSN MMMs has been proven to be an effective integral modification method [59,60,61]. The introduction of nanofillers effectively enhances the OSN performance of HF OSN membranes, but the defect-free preparation of MMMs is still challenging because the agglomeration of nanofillers is general during the preparing process of MMMs [21].

#### 4.1.3. Other Integral Treatment

In addition to crosslinking treatment and introducing nanomaterials to prepare MMMs, there are other treatment methods to treat HF OSN membranes, such as thermal treatment [32,48,60,73,77], acid [50,77]/alkali [50] treatment, and activation [68].

The thermal treatment has been used to accelerate the reaction [48], to meet the reaction conditions [32,60], and to dry membrane fibers [73,77]. Acid treatment is used to oxidize the substrate to make it hydrophilic [50,77] and alkali treatment is used to make the ceramics and polymers tightly bonded [50]. Activation refers to the exposing of membranes prepared to harsh solvents (e.g., DMF and DMSO) [68], and its concept was first reported by Prof. Livingston’s group [94]. After activation, the OSN membranes show greatly improved solvent permeance compared with the pristine membranes [12]. Goh et al. [68] did similar work in the field of HF OSN membranes and the results showed that the permeation performance of the activated HF OSN membranes was significantly improved, while the selectivity remained almost unchanged. 

### 4.2. Surface Modification

The physical (e.g., pore size, porosity, and roughness) and chemical properties (e.g., hydrophilic/hydrophobic, swellability, surface chemistry) of HF OSN membranes’ surface could significantly affect the performance and stability of the HF OSN membranes during OSN process [2,91,95]. Therefore, it is necessary to treat the membrane to meet the separation requirement. Some works have been published about the surface modification for preparing HF OSN membranes including IP, coating, introducing nanomaterials in the selective layer, etc. The discussion of IP methods is skipped because it has already been described in the Section 3.2.1.

#### 4.2.1. Coating

Coating is a commonly used method for surface modification of HF OSN membranes. The popular method is coating on the surface of ceramic [50,89,90] and polymer [18,19,39,62,77,87,90] substrates.

#### 4.2.2. Introducing Nanomaterials in the Selective Layer

One strategy to change the surface properties or morphology of the prepared membranes is the introduction of nanomaterials [15,59,60,61,67]. The work of Prof. Su [15] showed that the permeance of HF OSN membrane increased by 65% after doping with GO meanwhile maintaining the high solute rejection.

#### 4.2.3. Other Treatment Methods

There are other strategies to alter the surface properties and morphology of HF membranes, including vacuum filtration [96], CVD [32], immersing HF OSN membranes in solutions containing macrocyclic molecules [30], growing of COF layers in situ [28], thermal treatment [73], etc.

Table 4 shows the summarization of the treatment methods of HF OSN membranes.

## 5. Materials to Fabricate HF OSN Membranes

Many types of polymer materials have been used to fabricate HF OSN membranes, including PI [15,28], PAN [47,48], PBI [19,54], PA [52,61], PPTA [32], PVDF [18], polyetherimide [67,86], PAI [31,42], PPSU [43], PANi [44], and polypropylene [77]. PI is the most widely used polymer to prepare OSN membranes because of its good heat resistance, chemical stability, easy processing, etc. [92]. However, PI also has disadvantages, e.g., its tendency to hydrolyze [97], high cost [98,99], and low solvent permeance [28], which is unfavorable for its commercial application. PAN has the advantage of low cost, good heat, and solvent resistance [100] and the disadvantage of not being stable in NMP [101]. PBI has high mechanical strength and chemical stability, but PBI substrates is brittle [19]. Membranes made of ceramic could work under harsh conditions [102], but often require modification to meet the requirements of non-polar solvent separation [33,50,90,96].

Most polymer membranes would swell or dissolve when exposed to some organic solvents due to the interaction between the polymer and the solvents, resulting in loss of structural integrity and separation properties [27]. In addition, from a practical perspective, it is necessary to know the solubility of the polymers to be able to make dope solutions from the polymers. Therefore, it is necessary to understand the interaction between the polymer and the solvent. 

The interactions between polymer and solvent could be estimated based on their Hansen solubility parameters [103] (HSPs), *δ*, which could be considered as a function of the individual contributions related to dispersive forces (*δ_d_*), polar interactions (*δ_p_*), and hydrogen bonding (*δ_h_*) [104]. The smaller difference between the Hansen solubility parameters of the two substances, the stronger the interaction between them (i.e., the higher solubility) [105,106]. According to the recommendation of Buonomena et al. [107], the difference in Hansen solubility parameters between the solvent and the polymer could be calculated by the following equation:(3)$$\delta_{S-P}=\sqrt{{\left(\delta_{d,P}-\delta_{d,S}\right)}^{2}-{\left(\delta_{p,P}-\delta_{p,S}\right)}^{2}-{\left(\delta_{h,P}-\delta_{h,S}\right)}^{2}}$$
where *S* and *P* represent solvent and polymer, respectively, and $\delta_{S-P}$ represents the difference in solubility parameters between the solvent and the polymer

It should be noted that the HSP differences are only theoretical calculations and are for reference only [108]. The swelling of various polymer membranes in organic solvents was tested experimentally by Kappert et al. [109], and it could be concluded that the swelling phenomenon of polymer membranes cannot be explained only by the similarity of polymer–solvent structure and the polarity of the solvent.

There have been several works reporting a linear relationship between solvent properties (e.g., total HSPs, molar volume, and viscosity) and the pure solvent permeability of some polar organic solvents [48,87,110,111,112,113]. Therefore, the permeability of a specific membrane to some pure solvents could be predicted from the HSPs of the membrane, or the HSPs of a membrane could be calculated from the permeance test values of some pure solvents [114]. The total Hansen solubility parameter *δ_T,_**_S_* of the solvent could be calculated according to below [113].
(4)$$\delta_{T,S}=\sqrt{{\left(\delta_{d,S}\right)}^{2}+{\left(\delta_{p,S}\right)}^{2}+{\left(\delta_{h,S}\right)}^{2}}$$

Where *δ_T,_**_S_* represents the total Hansen solubility parameter of the solvent, and *δ_d,_**_S_*, *δ_p,_**_S_*, and *δ_h,_**_S_* are the Hansen parameters representing dispersive forces, polar interactions, and hydrogen bonding, respectively

The pure solvent permeance of some organic solvents the membrane could be predicted according to Equation (5) based on the solvent properties some other pure solvent permeances obtained from actual tests [113].
(5)$$P=K\times \frac{\delta_{T,S}}{\eta \cdot MV}+C$$
where *P* represents the pure solvent permeance, *K* and *C* are constants (calculated from the actual measured pure solvent permeances, and *δ_T,_**_S_*, *η*, and *MV* represent the total Hansen solubility, molar volume, and viscosity of organic solvent

It must be noted that the HSPs and calculated pure solvent permeabilities are theoretical calculations and for reference only, especially when calculating small molecules (small molecules such as methanol and acetone may give “anomalous results”) [50,115]. 

Table 5 shows the name, abbreviation, structure, and Hansen solubility parameters of the polymers to fabricate HF OSN membranes, and Table 6 shows the molecular masses (M), molar volumes (MV), boiling temperature (T_b_), densities (*ρ*, 25 °C), viscosities (*η*), and Hansen solubility parameters (*δ*) of some commonly used solvents.

## 6. Status Quo, Challenges, and Outlook of HF OSN Technologies

Over the past 20 years, the OSN performance of HF OSN membranes has been improving. HF OSN membranes can now achieve higher output and product purity at the same production cost. However, preparation difficulties and scale-up effects have limited the commercial application of HF OSN membranes [73,87], and there are still no commercially available HF OSN membranes [15,17,30,31,126]. In the following, a brief overview of the status quo, challenges, and future outlook of HF OSN membrane development is presented.

### 6.1. Module Comparasion

Compared with spiral-wound modules (SWMMs) made of FS membranes, HF modules are more suitable for commercialization due to its high packing density (up to 10 times higher than that of the SWMMs) [127,128], low fouling tendency [17], self-supporting structure [16], and simple module fabrication [129]. 

### 6.2. Sustainable and Green Preparation of HF OSN Membranes

Currently, the major membrane preparation processes often involve the use of many organic solvents and hazardous chemicals (especially amines, which are harmful to humans) and result in a large amount of wastewater containing organic solvents which are difficult to treat and recycle. From a sustainable and green point of view, the major membrane manufacturing processes are environmentally unfriendly [130]. In recent years, many published papers have reported on sustainable, environmentally friendly membrane materials and membrane preparation processes [52,61,131,132,133,134,135,136,137,138,139,140,141,142]. It is foreseeable that one of the common areas for future HF OSN membrane R&D should be the green and sustainable preparation of HF OSN membranes.

### 6.3. Developing New Membrane Materials

Developing novel membrane materials is an effective strategy to improve the performance of HF OSN membranes [2]. Researchers have explored many materials to prepare high performance OSN membranes and published many review articles summarizing the progress of the materials [34,143,144,145,146,147,148] and preparation [149] of OSN membranes. PBI, cellulose, block copolymers, COFs, materials with intrinsic structure (e.g., PIMs and conjugated microporous polymers), and carbon-based materials (e.g., carbon molecular sieves and carbon nanotubes) may be the next-generation OSN membrane materials because of their high permeability to certain solvents. In addition, braid-reinforced HF OSN membranes have shown great commercial potential due to their significantly improved mechanical strength.

### 6.4. Standardizedly Reporting HF OSN Membranes and OSN Process

Probably due to researchers’ personal preferences or experimental costs, different testing systems were adopted in the OSN performance tests of HF OSN membranes, including experimental configurations (e.g., cross-flow configuration [19] and dead-end configuration [67]), operating parameters (e.g., pressure [51,55,62,73] and feed flow rate [15,87]), solvents (e.g., EtOH [15], acetone [28], THF [19], etc.), solute, (e.g., dyes [18,54] and APIs [30,31]), etc. In addition, many reported works lack the reporting of some necessary information (e.g., reproducibility). The differences in testing systems and the lack of some necessary information cause many published works about HF OSN membranes report isolated data, which cannot be directly compared with data reported in other published works [150,151]. Therefore, detailed and standardized reporting the entire OSN process which the preparation, OSN performance testing [152] and characterization [150] of HF OSN membranes is important for the reliability, reproducibility, and comparability of the published works about HF OSN membranes. It is recommended to use a cross-flow configuration to test the OSN performance of HF OSN membranes and to report the whole processes (including the preparation, OSN performance testing and characterization of HF OSN membranes) in a complete and standard way [150], which is important for hollow fibers and other configurations equally.

### 6.5. From Lab Scale to Commercial Scale

Commercialization of HF OSN membranes requires the study of HF OSN membranes at an industrial scale. The common length of commercial HF components is 1–2 m (40–80 inches) with a diameter of 10–20 cm (4–8 inches) [17,153,154], while the HF modules used by academia are usually less than 40 cm in length and less than 25 cm (1 inch) in diameter [43,87]. The difference of module size makes a significant gap between HF OSN modules fabricated and tested at laboratory scale and those produced and operated at industrial scale, since a significant increase in length leads to significant changes in process parameters under similar operating conditions [153,155]. In addition, the properties of the fluid (e.g., density, viscosity) could also lead to variations in process parameters [153,156]. Thus, theoretically describing the fluid flow and mass transfer phenomena during HF OSN process and fabricating and testing large size HF OSN modules (>4 inches in diameter and >40 inches in length) is necessary for the correct prediction of the separation performance of commercial large size HF OSN membrane module. Modeling, module fabrication, and testing of HF OSN process in commercial dimensions should be accomplished to bridge the gap between lab-scale and commercial dimensions of HF OSN membranes.

### 6.6. Extending HF OSN’s Chemical Space

Currently, the number of solutes tested for OSN performance is still limited, and they are mainly APIs, dyes, and oligomers [150]. Only a tiny fraction of the vast number of molecules has been discovered so far [157]. The lack of diversity in the current OSN chemistry space may hinder the expansion of OSN applications and a deeper understanding of small molecule permeability in OSN process [152,158]. More work should be done on exploring various solutes, which could contribute to the expansion of the chemical space OSN and increasing the commercial values of OSN.

## 7. Conclusions

This review summarizes the research progress of HF OSN membranes, including membrane preparation, membrane materials, and membrane treatment. This review also discusses the challenges of the commercialization of HF OSN membranes and further predicts the future direction of the research and development of HF OSN membranes.

In order to speed up the development and commercialization of HF OSN membranes, future academic research work should focus on: (i) exploring sustainable and green preparation methods of high performance HF OSN membranes at low costs; (ii) detailed and standardized reporting the preparation, OSN performance testing, and characterization of HF OSN membranes; (iii) systematic and in-depth study of the mechanism of HF OSN membrane preparation and OSN processes; (iv) solving technical challenges related to commercialization of HF OSN membranes, such as large size modules preparation, membrane fouling and instability; (v) exploring more emerging OSN processes, such as high temperature OSN, fractionation, and precise molecular sieving; and (vi) exploring recycling and regeneration of waste membranes.

## Figures and Tables

**Figure 1 membranes-12-00995-f001:**
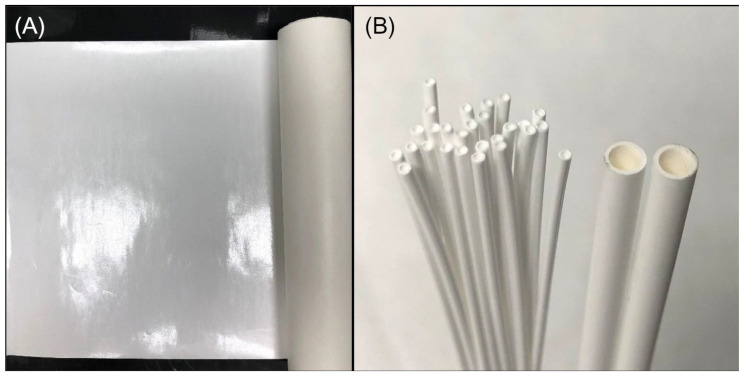
Photos of (**A**) FS and (**B**) hollow fiber (left)/tubular (right) membranes. Reprinted with permission from Ref. [20]. Copyright 2021, Elsevier.

**Figure 3 membranes-12-00995-f003:**
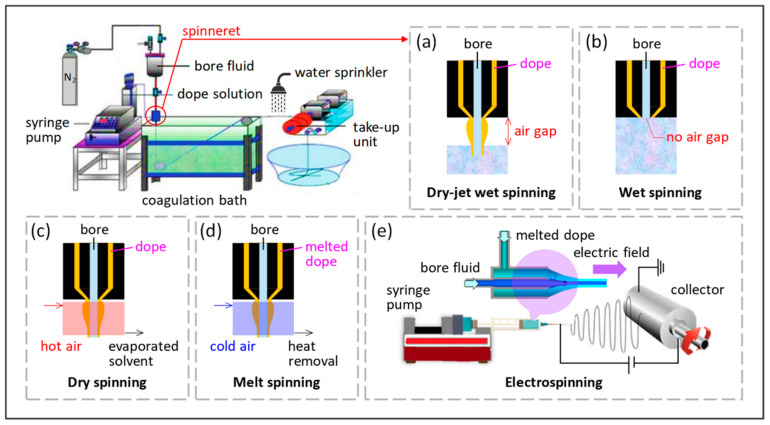
Schematic diagram of different hollow fiber spinning techniques including (**a**) dry-jet wet-spinning, (**b**) wet spinning, (**c**) dry spinning, (**d**) melt spinning, and (**e**) electrospinning. Reprinted from [16].

**Figure 4 membranes-12-00995-f004:**
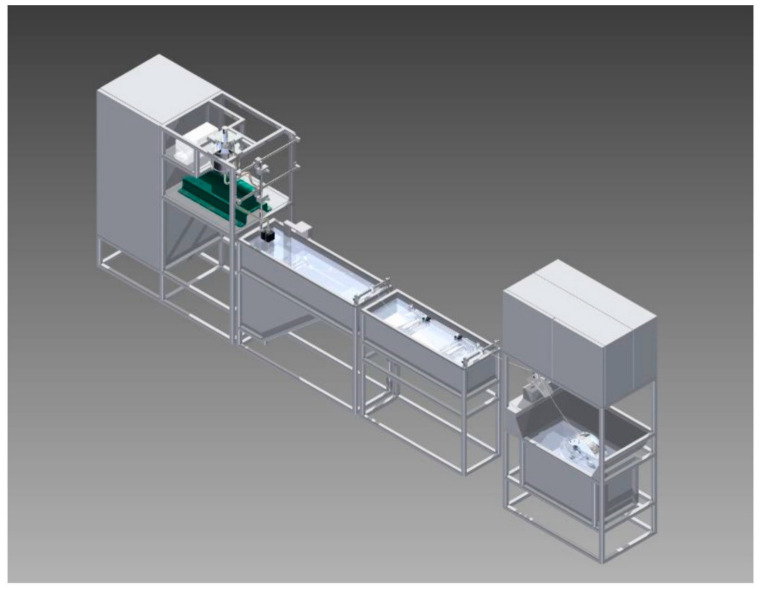
Schematic depiction of a hollow fiber membrane fabrication system. Reprinted from [40].

**Figure 5 membranes-12-00995-f005:**
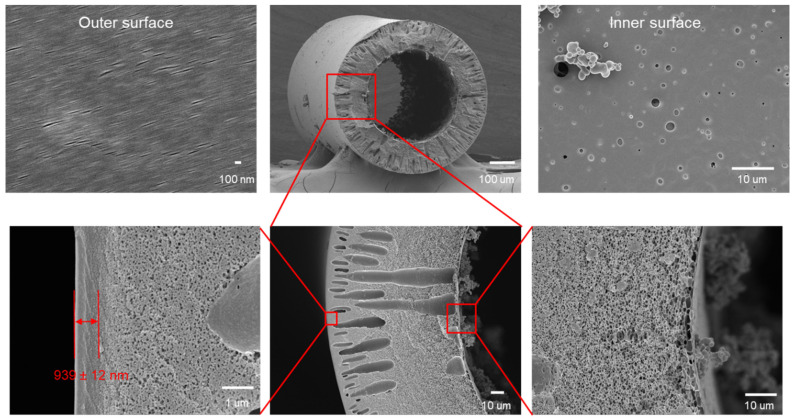
Surface and cross-sectional morphologies of a kind of ISA HF OSN membrane with a 939 nm thickness dense selective layer on its outer layer. Reprinted with permission from Ref. [31]. Copyright 2021, Elsevier.

**Figure 6 membranes-12-00995-f006:**
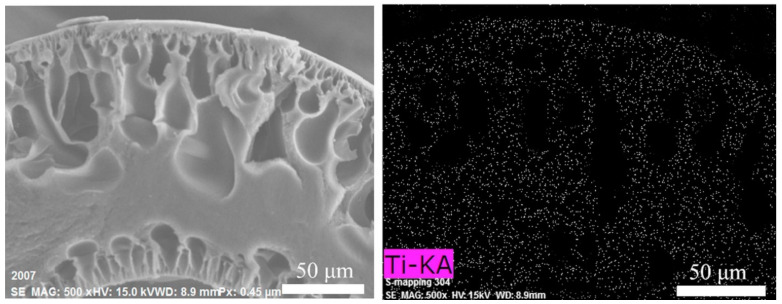
The SEM observation and EDS analysis of the cross-section of PAA HF MMMs, which shows the nanofillers are uniformly distributed in the HF cross-section. Reprinted with permission from Ref. [60], Copyright 2019, Elsevier.

**Figure 7 membranes-12-00995-f007:**
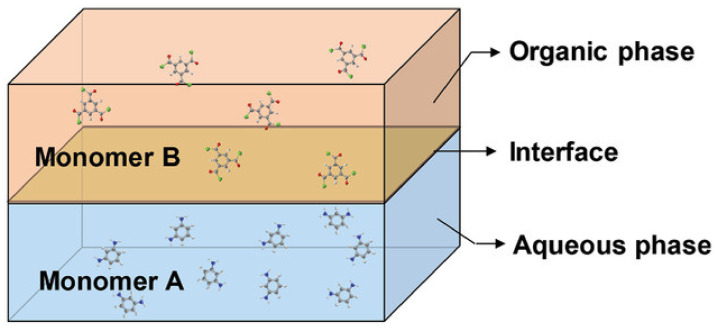
Schematic illustration of IP. Reprinted with permission from Ref. [34]. Copyright 2020, John Wiley and Sons.

**Figure 8 membranes-12-00995-f008:**
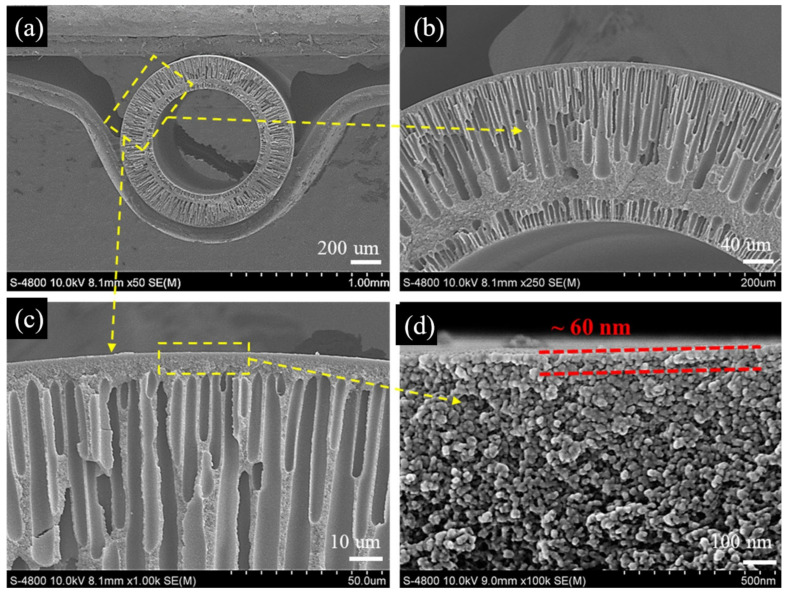
The SEM images of the cross section of PI TFC HF OSN membranes with a 60 nm thickness dense PA selective skin layer including (**a**) 50× magnification; (**b**) 250× magnification; (**c**) 1000× magnification; (**d**) 100,000× magnification. Reprinted and adapted with permission from Ref. [15]. Copyright 2022, Elsevier.

**Figure 9 membranes-12-00995-f009:**
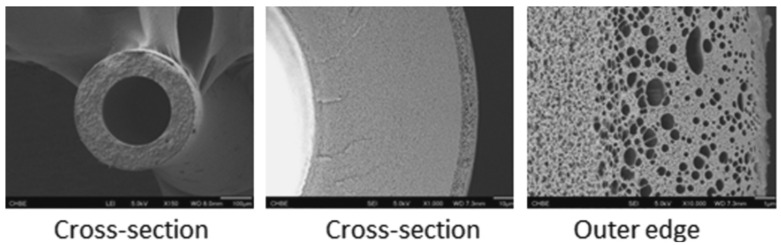
The morphology of a dual-layer HF membrane. Reprinted and adapted with permission from Ref. [73]. Copyright 2015, Elsevier.

**Figure 10 membranes-12-00995-f010:**
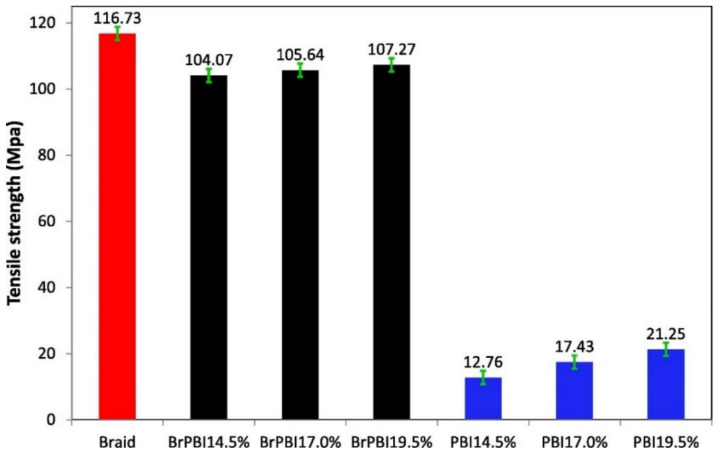
A comparison of tensile strength between braid-reinforced and self-support PBI hollow fibers. Reprinted with permission from Ref. [19]. Copyright 2022, Elsevier.

**Figure 11 membranes-12-00995-f011:**
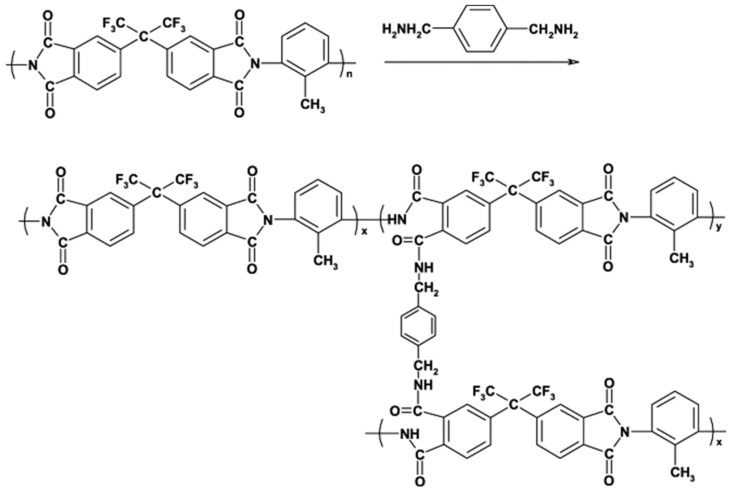
The mechanism of crosslinking between amines and PI. Reprinted with permission from Ref. [93]. Copyright 2003, Elsevier.

**Table 1 membranes-12-00995-t001:** OSN performances of ISA HF OSN membranes.

Membrane	Treatment Methods	Solute (MW, g mol^−1^)	Solvent	Pressure (bar)	Solvent Permeance (L m^−2^ h^−1^ bar^−1^)	Rejection (%)	Reference
PVDF	Triethylamine crosslinked and EGDGE secondly crosslinked	MO (327)	EtOH	1	0.70	∼98	[18]
PBI	K_2_S_2_O_8_ crosslinked	MB (320)	Acetone	10	∼3	∼99	[54]
PBI	K_2_S_2_O_8_ crosslinked	MB (320)	IPA	10	∼1	∼96	[54]
PBI	K_2_S_2_O_8_ crosslinked	MB (320)	EtOH	10	∼2	∼99	[54]
P84 PI	HDA crosslinked	RDB (479)	EtOH	10	1.83	98.0	[51]
PBI	DBX covalent crosslinked	TG (176, 5 wt. %)	Acetone	10	0.35	99.0	[30]
PBI	DBX covalent crosslinked	MO (327)	Acetone	10	1.85	77.5	[30]
PBI	DBX covalent crosslinked	RBB (626)	Acetone	10	1.95	99.5	[30]
PAI	TAEA crosslinked	MB (320)	MeOH	2	1.87	96.0	[31]
PAI	TAEA crosslinked	VbB (506)	MeOH	2	1.87	98.4	[31]
PAI	TAEA crosslinked	MB (320)	MeOH	2	1.00	99.7	[31]
PAI	TAEA crosslinked	VbB (506)	MeOH	2	1.87	99.9	[31]
Polyamide 6 MMM	Thermal annealing	Cyanocobalamin (1355)	MeOH	8	∼0.21	∼94	[61]
P84 PI	PEI coating, HDA crosslinked	RDB (479)	EtOH	10	1.20	81.0	[39]
P84 PI	PEI coating, GA and HDA crosslinked	RDB (479)	EtOH	10	0.45	96.5	[39]
PAN	Hydrazine monohydrate and tannic acid crosslinked	MB (320)	MeOH	3	1.28	71.6	[48]
PAN	Hydrazine monohydrate and tannic acid crosslinked	EB (961)	MeOH	3	1.28	100	[48]
PBI	Sulfuric acid protonated	TC (444)	MeOH	5	3.50	98	[50]
PIM-1 on PI substrates	p-xylylenediamine crosslinked	RB (1017)	EtOH	0.7	∼2	86	[62]
PMDA-ODA PI	Thermal treatment	FG (808)	DMF	10	2.50	90	[60]
Cellulose	–	CR (696)	EtOH	0.2	6.00	∼94	[55]
Polyamide 6	–	Vitamin B12 (1355)	MeOH	3	0.27	96.3	[52]
P84 PI	HDA crosslinked	BBR (826)	Acetone	5	3.98	99.9	[59]
P84 PI	HDA crosslinked	MB (320)	IPA	5	0.60	97.2	[59]
P84 PI MMM	HDA crosslinked	BBR (826)	Acetone	5	4.31	99.9	[59]
P84 PI MMM	HDA crosslinked	MB (320)	IPA	5	0.53	99.8	[59]
PAN	Hydrazine monohydrate crosslinked	RBB (626)	EtOH	2.8	2.32	99.9	[47]
PAI	APTMS crosslink	RB (1017)	IPA	2	6.4	97	[42]
PANi	Thermal crosslinked and with doping in acids	Oligostyrene (500)	Acetone	6	1.53	∼95	[44]

**Table 2 membranes-12-00995-t002:** OSN performances of composite HF OSN membranes.

Membrane	Treatment Methods	Solute (MW, g mol^−1^)	Solvent	Pressure (bar)	Solvent Permeance (L m^−2^ h^−1^ bar^−1^)	Rejection (%)	Reference
PEI-isophthaloyl dichloride on PP substrates	IP	BBR (826)	MeOH	4.13	1.47	88	[77]
PBI-PET dual layer	K_2_S_2_O_8_ crosslinked	RB (1017)	MeOH	10	3.60	99.5	[19]
MPD-TMC on PI substrates	IP	RBB (626)	MeOH	16	0.90	99.3	[73]
PEI/PIP-TMC on PI substrates	IP	RB (1017)	Acetone	2	11.6	99.9	[76]
PEI/PIP-TMC on PI substrates	IP	AF (585)	IPA	2	4.5	91.8	[76]
MPD-TMC on PI substrates	IP	AF (585)	Acetone	2	24.2	99.4	[68]
MPD-TMC on PI substrates	IP	MO (327)	EtOH	2	2.33	98.6	[68]
MPD-TMC on PI substrates	IP	Mr (269)	MeCN	2	10.58	90.1	[68]
MPD-TMC on PI substrates	IP	Levofloxacin (361)	MeCN	2	10.58	98.2	[68]
P84-PET dual layer	HDA crosslinked	Ch (515)	DMF	6	∼2	98.0	[87]
Hydrophobic COF on PI substrates	IP	L-α-lecithin (758)	Hexane	1	266.27	∼100	[28]
Hydrophobic COF on PI substrates	IP	FG (808)	Acetone	1	395.21	98.9	[28]
Hydrophobic COF on PI substrates	IP	RB (1017)	EtOH	1	98.44	92	[28]
Hydrophobic COF on PI substrates	IP	RB (1017)	IPA	1	61.68	99.1	[28]
Hydrophobic COF on PI substrates	IP	FG (808)	IPA	1	61.68	99.5	[28]
MPD-TMC on PI substrates	IP	RDB (479)	EtOH	5	1.20	100	[15]
Go doped MPD-TMC on PI substrates	IP	RB (1017)	MeOH	5	5.80	99	[15]
Go doped MPD-TMC on PI substrates	IP	RDB (479)	EtOH	5	1.99	100	[15]
TiO2@GO-MPD-TMC on PAN/ceramic substrates	IP	BTB (624)	EtOH	8	∼4.1	∼95	[50]
GO on PTEI substrates	Coating	RB (1017)	Acetone	1	4.0	91	[67]
PI/polyetherimide dual layer	HDA crosslinked	TC (444)	MeOH	Not mentioned	3.70	>99	[86]
polypyrrole on PPTA substrates	CVD	CR (696)	DMAc	6	1.17	99.3	[32]
polypyrrole on PPTA substrates	CVD	Eosin Y (648)	DMAc	6	1.10	93.4	[32]
polypyrrole on PPTA substrates	CVD	MB (320)	DMAc	6	1.04	84.5	[32]
polypyrrole on PPTA substrates	CVD	CR (696)	EtOH	6	1.64	99.5	[32]
PBI/P84 dual layer	HPEI crosslinked	MB (320)	MeCN	1	1.58	99.5	[88]
PBI/P84 dual layer	HPEI crosslinked	MB (320)	MeOH	1	2.60	99.1	[88]

**Table 3 membranes-12-00995-t003:** Advantages and disadvantages of various crosslinking methods. Reprinted and adapted with permission from Ref. [49]. Copyright 2019, ACS Publications.

Polymer	Crosslinker	Advantages	Disadvantages
PBI	K_2_S_2_O_8_	Green	Long crosslinking time
PBI	DBX	Highly stable, high permeance, low MWCO	High temperature modification
PI	Thermal	High selectivity, inexpensive	Low permeance
PI	Chemical (amines)	Adjustable performance by varying the crosslinker	Long crosslinking time, relatively toxic
PAI	APTMS	Improved hydrophilicity and tensile modulus	Brittle
PAN	Hydrazine	Inexpensive, room-temperature modification	High swelling in NMP, DMSO; toxic
PANi	Thermal	Inexpensive, short modification time	Acid dopants required, high temperature modification

**Table 4 membranes-12-00995-t004:** Treatment methods of HF OSN membranes.

Method	Advantages	Disadvantages
Crosslinking	Improved resistant to solvents and plasticization;	Adding extra steps; some crosslinkers toxic
Introducing nanomaterials	Improved OSN performance and mechanical strength	Defects due to particle agglomeration
IP	Thin selective layer with good OSN performance	Membrane fouling; hard to remove membrane fouling; skin layer shedding
Acid/alkali treatment	Improved surface chemistry	Adding extra steps; long treating time
Thermal	Low cost	Low permeance
Coating, CVD, and vacuum filtration	Changed surface properties	Difficulties to make the coating even
Growing COFs in situ	Significantly higher permeances of non-polar solvents	Long reaction time
Soaking membranes in polyol (e.g., 50/50 wt. % glycerol/water solution)	Avoiding the pore collapse	Long soaking time

**Table 5 membranes-12-00995-t005:** The name, abbreviation, HSPs, and structure of the polymers to fabricate HF OSN membranes.

Polymer	Structure	Reference ^a^	Hansen Solubility Parameters (MPa^0.5^)	Reference ^b^
δ_d_	δ_p_	δ_h_
P84 polyimide (PI)	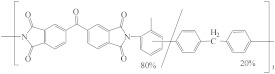	[15,39,45,51,59,68,76,86,87,88]	20.4	20.4	10.3	[116]
Matrimid^®^ 5218 polyimide (PI)	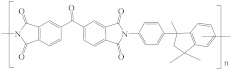	[28,62,73]	18.7	9.5	6.7	[117]
Polybenzimidazole (PBI)	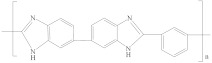	[19,30,50,54,88]	17.3	8.7	8.9	[118]
Polyacrylonitrile (PAN)	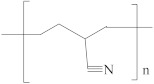	[47,48,50]	23.3	15.5	11.4	[119]
Polyaniline (PANi)	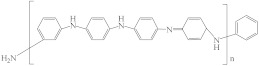	[44]	25.1	4.2	7.4	[120]
Polyphenylsulfone (PPSU)	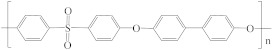	[43]	18.7	5,0	7.4	[121]
Torlon^®^ 4000T-MV Polyamide-imide (PAI)	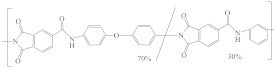	[31,42]	21.3	11.84	7.18	[122]

^a^: the references that report the use of this polymer for the preparation of HF OSN membranes; ^b^: the papers reporting the HSPs.

**Table 6 membranes-12-00995-t006:** Molecular masses (M), molar volumes (MV), boiling temperature (T_b_), densities (*ρ*^25 °C^), viscosities (*η*^25 °C^), and Hansen solubility parameters (*δ*) of some commonly used solvents and non-solvents [123,124]. Reprinted and adapted with permission from Ref. [125]. Copyright 2009, Elsevier.

Solvent	M (g/mol)	MV (cm^3^/mol)	T_b_ (°C)	*ρ*^25 °C^ (g/cm^3^)	*η*^25 °C^ (cP)	Hansen Solubility Parameters (MPa^0.5^)		
						*δ* _d_	*δ* _p_	*δ* _h_
NMP	99.1	96.5	202.0	1.026	1.666	18.0	12.3	7.2
DMF	73.1	77.0	153.0	0.944	0.802	17.4	13.7	11.3
DMAc	87.1	92.5	165.0	0.936	0.927	16.8	11.5	10.2
THF	72.1	81.7	64.5	0.881	0.460	16.8	5.7	8.0
H_2_O	18.0	18.0	100.0	0.997	0.890	15.6	16.0	42.3
MeOH	32.0	40.7	64.6	0.787	0.551	15.1	12.3	22.3
EtOH	46.1	58.5	78.5	0.785	1.083	15.8	8.8	19.4
n-Propanol	60.1	75.2	97.1	0.800	1.943	16.0	6.8	17.4
IPA	60.1	76.8	82.0	0.781	2.044	15.8	6.1	16.4
1-hexanol	102.2	125.2	156.4	0.815	4.592	15.8	4.3	13.5
Acetone	58.1	74.0	56.0	0.784	0.303	15.5	10.4	7.0

## Data Availability

No new data were created or analyzed in this study. Data sharing is not applicable to this article.

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
