# Peer review of "Hollow Fiber Membrane for Organic Solvent Nanofiltration: A Mini Review"

_membranes, 2022, doi:10.3390/membranes12100995_

Round 1
Reviewer 1 Report
The manuscript presents a ‘review article’ on an important and timely topic. Such review does not exist yet. It fits well the scope of the journal. However, there are several important works missing, the depth of the discussions need to be improved, and some original figures added. This reviewer recommends major revision to carefully address the comments below before further consideration by the journal.
1, The authors should provide a reference for the organic solvent ‘reuse rate is less than 50%’ in the main text.
2, A table of figure should be added that compares certain OSN membranes with respect to their preparation and performance. An authoritative review article should have a detailed literature assessment rather than compilation of information. In this case, for instance, select PBI as polymer, and compare the preparation and OSN performance of all PBI membranes made in flat sheet, tubular and hollow fiber configurations. The used solvents, the used crosslinkers, the greenness of the methods, the surface modifications, the flux ranges, the MWCO ranges, the tested solvents, tested temperatures etc should be included in the table. It will be useful for the community to be able to directly compare how the membrane type (flat sheet versus hollow fiber) results compare.
3, The authors write that OSN membranes have an average diameter of 1 – 2 nm and molecular weight cut-off (MWCO) of 100 – 1000 Da. There are two problems with this statement. The membranes do not have an average diameter of 1-2 nm. Most likely they wanted to write that the pore size of the OSN membranes are about 1-2 nm. However, the actual pore size in OSN can be even smaller, in the range of 0.2 nm. The MWCO range was also expanding and now not only the 100-1000 but 50-2000 Da range is used for separations by OSN. The reference for this sentence (#8) is a specific research article, which seems to be inappropriate for such a broad statement, instead refer to the review you are already citing under #3.
4, Section 2.1 on processes should also mention fractionation and high temperature OSN with examples as these are both important emerging areas in OSN with recent but scarce literature.
5, Bridging the gap between lab-scale and commercial dimensions of hollow fiber nanofiltration membranes should be mentioned as scale-up is an important aspect (10.1016/j.memsci.2021.119100).
6, Table 3 reveals the polymers that were used to fabricate hollow fiber OSN membranes. This table should have more information, more columns, on the main properties and characteristics of these polymers, and the resulting membranes. For instance, what water contact angle, what polarity ranges, these polymers can give.
7, More critical discussions should be included in the manuscript. The review article should go beyond compilation of literature and it should have a critical edge. Each section should discuss pros and cons, and add a critical evaluation and thoughts on the literature presented.
8, The membrane structure given in lines 35-36 is incomplete as carbon molecular sieves as emerging structural types should be also mentioned (10.1016/j.apmt.2022.101541).
9, Fabrication of microporous polyamide selective layer on macroporous ceramic hollow fibers via direct interfacial polymerization should be mentioned, as in general there is not much discussion on ceramics in the manuscript (10.1016/j.memsci.2022.120710).
10, From a practical perspective, it is necessary to know the solubility of the polymers to be able to make dope solutions from the polymers but at the same time it is also necessary to know what are the solvents in which the polymers are insoluble and the membranes stable so that OSN can be performed without surface modification or post-treatment. Solubility information for all polymers mentioned should be added in the manuscript.
11, The application examples (line 29) should include biorefineries (10.1002/cssc.201802747), natural product isolation (10.1021/acssuschemeng.9b04245), organocatalysis (10.1021/acscatal.8b01706), solvent exchange (10.1002/ange.201607795).
12, The options for hollow fiber surface modification and post-modification should be added. A figure presenting the different methodologies available, their use and the results, their comparison for hollow fiber OSN should be added.
13, The scale-up of hollow fiber membranes should be discussed in a short subsection. What are the pros and cons of hollow fiber membranes for OSN compared to other types? What is the scale of product and size of hollow fiber modules? Which are the commercially available hollow fiber OSN membranes? The authors should give more information on these aspects in the review article.
14, There is a previous review on hollow fiber membranes in the same journal, which should be acknowledged, and distinguished from the current review which focuses on OSN while the previous one on NF in general (10.3390/membranes11110890).
15, The authors should briefly mention the need for standardized reporting in the OSN field, which is important for hollow fibers and other configurations equally (10.1039/D0GC00775G).
16, A general schematic for the preparation of hollow fiber membranes should be added for the readers who are not familiar with this technique.
17, The section 2.4 on performance matrix mentioned the advantages of dyes for separations. Indeed dyes have their own advantages and disadvantages among other solute markers, and therefore refer to 10.1016/j.memsci.2021.119929 on the diversity requirements for OSN markers.
18, The conclusion and outlook section should be more elaborate. In its current form this section is too short and vague. Mention specific outcomes, results of data comparison, all the information that is not obvious from reading individual articles. The outlook should include the authors’ expert opinion on the topic, and where impact is expected, what the hot areas are.
Reviewer 2 Report
This article presented an extensive review on hollow fiber membranes for organic solvent nanofiltration, which covered the most research progress in this field and summarized in detail about membrane materials, membrane types, fabrication methods and membrane performance. As such, this work is of interest to the membrane community and is publishable in Membranes journal. However, this manuscript suffers from several issues, the referee thinks that this work should significantly improve before it can publish. Comments are below:
1. As presented, a lot of active voice is used in this manuscript, the writing is not acceptable for the scholarly journal, and there are problems with sentence structure and clause construction. This manuscript needs careful editing by someone with expertise in technical English editing paying particular attention to English grammar, spelling and sentence structure.
2. A schematic diagram of membrane types should be provided in the 3 section according to the different membrane structures so that the membrane types are clear to the reader.
3. The equation of solvent permeation flux or permeance should be added in the 2.4 section.
4. Although Table 1 appeared in the main text, there was not any discussion. Please give some comments or discussions for Table 1.
5. Line442, “Fig. shows…”, which Figure? Line 171, the “(>99)” lacks the unit.
6. Some typical figures should be selected from previous literature for every section, such as the interfacial polymerization process and crosslinking treatment process, etc.
7. The 3 section included membrane materials, membrane type, and membrane preparation, so the title of the 3 section is not suitable. The headings at the same level should have a parallel structure.
Round 2
Reviewer 1 Report
THe manuscript has improved and a significant portion has been updated.